# A deep hybrid learning pipeline for accurate diagnosis of ovarian cancer based on nuclear morphology

Duhita Sengupta[1,2☯], Sk Nishan Ali[3☯], Aditya Bhattacharya[3], Joy Mustafi[3], Asima Mukhopadhyay[4¤a¤b], Kaushik Sengupta[1]*

1 Biophysics and Structural Genomics Division, Saha Institute of Nuclear Physics, Kolkata, West Bengal, India, 2 Homi Bhaba National Institute, Mumbai, India, 3 Artificial Intelligence and Machine Learning Division, MUST Research Trust, Hyderabad, Telangana, India, 4 Chittaranjan National Cancer Institute, Newtown, Kolkata, West Bengal, India

☯ These authors contributed equally to this work.
¤a Current address: Northern Gynaecological Oncology Centre, Queen Elizabeth Hospital, Gateshead, United Kingdom
¤b Current address: Formerly at Tata Medical Center, Kolkata, West Bengal, India
* kaushik.sengupta@saha.ac.in

**Data Availability Statement:** All relevant data are within the paper and its Supporting Information files.

## Abstract

Nuclear morphological features are potent determining factors for clinical diagnostic approaches adopted by pathologists to analyze the malignant potential of cancer cells. Considering the structural alteration of the nucleus in cancer cells, various groups have developed machine learning techniques based on variation in nuclear morphometric information like nuclear shape, size, nucleus-cytoplasm ratio and various non-parametric methods like deep learning have also been tested for analyzing immunohistochemistry images of tissue samples for diagnosing various cancers. We aim to correlate the morphometric features of the nucleus along with the distribution of nuclear lamin proteins with classical machine learning to differentiate between normal and ovarian cancer tissues. It has already been elucidated that in ovarian cancer, the extent of alteration in nuclear shape and morphology can modulate genetic changes and thus can be utilized to predict the outcome of low to a high form of serous carcinoma. In this work, we have performed exhaustive imaging of ovarian cancer versus normal tissue and developed a dual pipeline architecture that combines the matrices of morphometric parameters with deep learning techniques of auto feature extraction from pre-processed images. This novel Deep Hybrid Learning model, though derived from classical machine learning algorithms and standard CNN, showed a training and validation AUC score of 0.99 whereas the test AUC score turned out to be 1.00. The improved feature engineering enabled us to differentiate between cancerous and non-cancerous samples successfully from this pilot study.

## Introduction

Ovarian malignancy is the 8th leading cause of cancer mortality among women, the 7th leading cause of cancer diagnosis worldwide, and the 3rd most common women's cancer in India

**Funding:** The author(s) received no specific funding for this work.

**Competing interests:** The authors have declared that no competing interests exist.

[1]. GLOBOCAN predicts a 56% increase in ovarian cancer incidence worldwide by 2050- and the majority (75%) of cancers are still diagnosed in late stages [2]. Multiple studies have shed light on the close association between lamin proteins and different types and classes of ovarian cancer. Capochichi et al. have elucidated a reduced expression of lamin A protein in epithelial ovarian carcinoma due to its degradation of the protein by caspase 6 in ovarian cancer cells [3]. Interestingly enough, contradictory results emerged from comparative proteomic analysis and immunohistochemistry analyses where lamin A expression was shown to increase in advanced stages of ovarian carcinoma [4]. Furthermore, the scenario gets even more complicated by the heterogeneous expression of lamin A/C within a population of tumor cells. Morphological hallmarks of ovarian cancer cells with reduced lamin A/C level included gross nuclear size aberration and onset of aneuploidy [5]. Nuclear lamins are type V intermediate filament proteins which form a thick meshwork or lamina underneath the inner nuclear membrane (INM) thus imparting proper size, shape, and mechanical rigidity to the nucleus [6]. Lamins also provide a scaffold for the binding of several proteins and chromatin. Lamins are associated with a wide range of nuclear functions like nuclear stability [7], genome organization [8], protein interaction [9], DNA damage repair [10], intracellular signaling [11], and play vital roles in replication [12], transcription [13], and splicing [14] as well. Lamins are mainly of A and B types which are coded by LMNA, LMNB1/B2 genes respectively. In many tumor types, lamin A/C levels are found to be elevated which in turn is associated with their aggressive metastatic potential, which cannot be explained by retro-differentiation [15]. Rather, increased levels of lamin A/C might play vital roles in tumor progression by helping the cells overcome the mechanical stress. It may also help in the recruitment of DNA damage repair proteins and thus resists DNA damage-induced cell cycle arrest. In either case, change in expression levels of lamin A/C largely remains a reliable prognostic marker for different tumor types and stages [16]. Being the major architectural protein of animal cell nuclei, lamins must be playing a vital role in this alteration of size [17]. Loss of lamin or their mutations leading to deformed nuclear morphology has been widely studied by different groups [6, 18, 19]. Interestingly, the differential expression pattern of lamins in different cancers has also been documented from research across the world [20].

Recent findings have demonstrated methods to study nuclear morphologies in the light of fluorescent imaging and deep learning [21]. The application of convolutional neural network pipelines in cancer diagnosis and detection has already been widely recognized and reported by different groups utilizing different technical and biological parameters. Currently, the rise in life expectancy is a global concern that is widely being triggered by an increase in incidences of age-related gynecological cancers [22]. Detection of cancer from histopathology images and immunocytochemical staining is widely being implemented worldwide [23, 24]. On the other hand, alterations in nuclear morphology have been acclaimed as hallmarks in various cancers and are used majorly in pathology or diagnostic purposes by clinicians to verify the degree of malignancy [25]. But these verifications based on manual observations are often cumbersome and sometimes prone to error.Various parametric and non-parametric methods are already in use for the diagnosis of various cancers [26, 27]. Several groups have reported deep learning techniques to classify different stages or subtypes of ovarian cancer based on different techniques and biological features [28, 29]. These rely on different classifiers based on differential features due to a shift from the familiar genomic or proteomic sketch of the non-cancerous counterpart. There are several methods to study alterations in various nuclear matrix proteins which are already reported to be used for cancer diagnosis like BLCA4 (Bladder and urothelial carcinoma protein 4) in bladder cancer [30], AR-V7 (Androgen Receptor splice variant 7) in prostate cancer [31] and NMP 179 (Nuclear Matrix Protein 179) in cervical cancer [32]. But these data were essentially non-parametric or in other words, not based on specific

annotations of prognostic markers. Morphometric studies in the context of cancer have been thus far evaluated in only a few other carcinomas but not in great detail in ovarian cancer.

Classical supervised machine learning algorithms have proved to produce great results with many limited data as compared to Deep Neural Network (DNN) based approaches, especially when we go for techniques like Boosting [33] and Bagging [34]. This assumption led us to form our intuition that a Deep Hybrid Learner (DHL) will use an Encoder network to extract low dimensional feature encodings from high dimensional raw images, and these encoded features are later used by an ensemble learning-based classical machine learning algorithm to produce the final classification result. Based on this hypothesis, we have performed a pilot study where we have utilized lamin-induced morphological changes of the nuclei as an important input parameter for developing an advanced deep hybrid learning (DHL) architecture. This is based on the calculated morphometric parameters as well as auto feature extraction from preprocessed images by deep learning techniques. For the first time, we have introduced the use of lamin A and B as specific markers for the classification of ovarian cancer. This in turn can aid in pathological inspections for diagnosing ovarian cancer. Systematic analyses of cancerous and non-cancerous samples were accomplished by confocal imaging. The images were denoised, sharpened, greyscaled, normalized, and augmented by a combination of operations like rotation, zoom, height shift, horizontal flip, etc. before using as inputs in a 21 layered CNN. The outputs were flattened and passed to classical machine learning algorithms. Morphometric parameters (Area, Perimeter, Circularity, Eccentricity, Loop Length, Foci Distance, Maximum Curvature, Normalised Curvature) of cancer and normal nuclei were taken as inputs in a parallel pipeline feeding into the same architecture. The decision function was generated based on the maximum probability score between Adaptive boosting and deep learning. The pattern of Learning curves, loss function, ROC-AUC, AUC-PR curves indicate the robustness of the deep hybrid learning model with very low overfitting.

## Materials and methods

### Tissue sample collection

Formalin-fixed paraffin-embedded tissues from ovarian cancer patients were obtained during frontline surgery at Tata Medical Center (TMC), Kolkata. We have obtained written consent in the form of material transfer and research agreement from Tata Medical Center/Tata Translational Cancer Research Center Biobank for the Pilot Study on the regulation of lamin proteins in High-Grade Serous Epithelial Ovarian Cancer (HGSC lamin study) IRB Ref No.: EC/TMC/45/15

### Immunohistochemistry

The paraffin-embedded blocks were cut into 4–5 μm sections and affixed onto glass slides. For removal of paraffin, the slides were immersed in xylene (3*10mins) followed by immersion in graded ethanol twice for 10 minutes in each (100%,95%,80%,70%), washed in ddH$_2$O twice for 5 minutes each. The tissues were immersed in 10mM sodium citrate buffer (pH 6), placed in a microwavable container, and heated at full power for 3 minutes and 80% power for the subsequent 12 minutes. After which the slide was allowed to cool gradually in the same buffer, followed by rinsing with ddH$_2$O twice for 5 minutes each followed by two rounds of wash in PBS for 5 minutes each. The rest of the immunocytochemistry procedure is mentioned in a previous article from our lab [35]. Primary antibody dilutions for Rabbit Anti Lamin A antibody [36, 37] (Sigma Aldrich L1293), Goat Anti Lamin B [38] (Santacruz sc-6217) antibody were 1:100 and 1:50 respectively. Secondary antibodies were conjugated with Alexa Fluor 488 (Green Fluorescence) and used at a dilution of 1:400.

## Image analysis and data presentation

Images were analyzed using ImageJ software (ImageJ bundled with 64-bit Java 1.8.0_112) [39]. Considering each nucleus as an ellipse, the equations used in the referred article [40] were followed to derive the values of the morphometric parameters like area, perimeter, loop length, circularity, eccentricity, foci distance, maximum curvature, normalized maximum curvature. Histograms were generated using the ROOT data analysis framework (Version 6, Release 6.08/ 06-2017-03-02) [41] which is an object-oriented C++ framework for large data storage, presentation, visualization, and statistical analysis. Each field from every tissue sample contained around 100 nuclei approximately. The length of the major and minor axes of each of the nuclei was measured manually by ImageJ. Mean, Standard error of mean and Standard deviation for the analysis of each parameter have been mentioned in the figure legends.

## Pre-processing and standard scaling

Upon qualitative visual inspection, we found that the stained cells from the original images are sometimes blurry due to non-uniform lighting effects. Hence, we hypothesized the need of having a custom image processing procedure. This would minimize these noisy effects and help the deep learning models to learn the key features effectively for the final classification. So, we have analyzed and compared the statistical properties of our dataset and with the ImageNet data. As fundamentally our images are different from ImageNet images, the raw RGB image quality was evaluated initially before going for pre-processing, to check how the images are different from ImageNet images (**S1 Table**). Although Deep Learning algorithms are believed to apply auto feature extraction methods, we wanted to additionally transform the raw data and make it easier for the model to unravel key features. Using morphological elliptical image filters, morphological masks were formed around the nuclei, and then using Gaussian Blur filter and pixel weights addition, the sharpened version of the images were obtained. Then the images were grey-scaled and normalized before feeding them into the deep learning model. The morphometric parameters used in this study have different units thus the input variables are of various scales. So, standardization of the scale is required to avoid generalization errors. Standard scaling of the data includes rescaling the distribution values by subtracting the mean and then dividing by standard deviation so that the distribution shifts to a mean of 0 and a standard deviation of 1 [42].

## Details of adaptive boost over morphometric data

Following standard scaling of the morphometric data and splitting the dataset into 75:25 training: test ratio, three classifiers (Adaptive Boosting Classifier, Random Forest Classifier [43], and Decision tree Classifier [44]) was used to generate probability scores to predict the data as cancerous or non-cancerous. Probability scores from the Adaptive Boosting Classifier were used as final outputs from Pipeline 1 [45].

## Controlled data augmentation

We used an image data generator to create millions of augmented images of a high perceptual quality that combine the properties and appearance of a different image, resulting in increased overall training efficiency [46]. A combination of rotation, zoom, width shift, height shift, horizontal flip, the vertical flip was implemented to augment the training dataset on the fly during the training of neural network. This resulted in a robust, reliable, generalized outcome. To further minimize any effect of overfitting due to our small dataset, we have applied controlled data augmentation techniques like slight rotation by 40 degrees, minimal zooming by 0.2%,

slight width, and height shift by a factor of 0.2, horizontal and vertical mirroring. This essentially helped us to virtually expand our training dataset and further reduce overfitting effects. The controlled data augmentation is applied for all the experimentation approaches mentioned.

## Deep hybrid learning model architecture

The entire Deep Hybrid Learning architecture used for this research work started with the pre-processing layer, and then the pre-processed images were passed through the model input layer. Deep Hybrid Learner utilized a Deep Convolutional Neural Network Layer for feature extraction. In our research, we have used a 21 Layered CNN which was inspired from Inception Net v3 [47] architecture. The CNN part consisted of a series of Incept layers and Squeeze Layers which are similar to grouped convolution layers with specific hyper-parameters and the nested Conv2D layer as illustrated in **Fig 1**. Input from image morphometric parameters are fed into the classifier through pipeline 1 as discussed previously. Details of each layer are illustrated in the network diagram provided in **S1 Fig**. The overall scheme of the architecture used in this project is depicted in **Fig 1**.

## Comparison of DHL with other deep learning approaches

In this research work, we have compared the efficacy of Deep Hybrid Learning with both XGBoost [33] and Random Forest variant [43], with a conventional Deep Neural Network (without transfer learning and having the same 21 layered CNN as DHL), DenseNet201 [48]

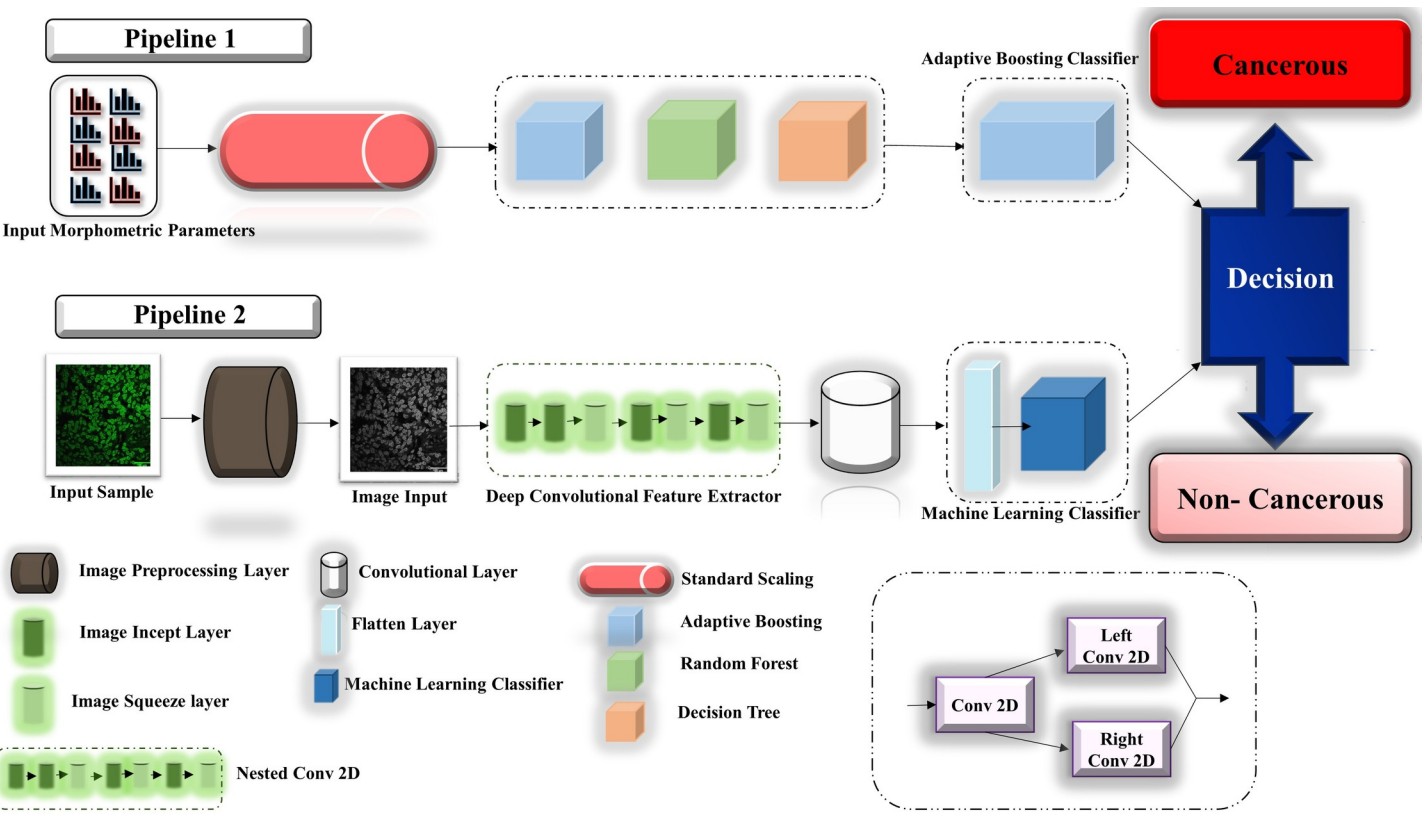

**Fig 1. The overall scheme of the algorithm.**

with transfer learning [49], InceptionNet v3 [47] with transfer learning [49], ResNet50 [50] with transfer learning [49] and VGG16 [51] with transfer learning [49].

## Decision function based on stacking of deep hybrid network and adaptive boosting

Probability scores from adaptive Boosting and Machine Learning classifiers were derived and the Decision function was based on the Maximum probability score of cancerous nuclei predicted correctly using Adaptive Boosting and that of cancerous nuclei predicted correctly using deep learning.

**Decision = Max (Prob (Cancerous nuclei Predicted Correctly using Adaptive Boosting), Prob (Cancerous nuclei Predicted Correctly using Deep Learning))**

## Results and discussion

### Distinct nuclear morphology of ovarian cancer tissues

FFPE blocks of normal and diseased (ovarian cancer) tissues were obtained from Tata Medical Center following the ethical guidelines. One part of the tissue samples were classified by the pathologists to be cancerous and non-cancerous by independent, unbiased methods and regular standards. Another part was stained with lamin A, and lamin B following proper antigen retrieval technique and imaged under the confocal microscope. One representative image from each of the tissue types has been shown in **Fig 2**. A visibly prominent enlargement of the cancer nuclei was observed with respect to the normal nuclei in both lamin A and lamin B-stained tissues.

### Morphometric classification, standard scaling, and adaptive boosting

Careful investigation revealed that the hallmark of the diseased tissues was characterized by prominent nuclear enlargement as reported earlier [52]. We quantified these changes as mentioned previously. With these sets of images, a gross morphometric analysis was performed based on the distribution of lamin A and lamin B proteins in the nucleus. Histograms were generated for each of the parameters using the ROOT data analysis framework [41] where the X-axis denotes the normalized number of nuclei with respect to the total number of nuclei calculated corresponding to the defined parameter and Y-axis denotes the measure of the parameter. It was evident from the plots (**Fig 3**), that the perimeter of most of the cancer nuclei from the total population exhibited an increase of 55–62% compared to most of the normal nuclei for both lamin A and lamin B-stained tissues (**Fig 3A and 3B**). A similar phenomenon was observed while measuring the area, where the area of most of the cancer nuclei was more than twice the area of most of the normal nuclei in the population (**Fig 3C and 3D**). Both the observations indicated an increase in the size of the cancerous nuclei. However, in the cancer nuclei, around 3% and 12% shifts from the normal were observed in the circularity and eccentricity values respectively which was not that significant denoting no prominent change in the shape (**S2A1, S2A2, S2B1, S2B2 Fig**). Eccentricity is a focal length (Distance from the center to one focus) and semi-major axis dependent variable. Still, to further validate, foci distance (2*Focal length) was also measured where the shift associated with eccentricity was supposed to get doubled according to the formulae. We could find a small increase in the Foci distance values of the cancer nuclei in comparison to the normal nuclei, which denotes an increase in the distance between the foci thereby approaching an elliptical nature (**S2C1 and S2C2 Fig**). Another common parameter in ellipse geometry is loop length, which is a focal length dependent variable, hence a rise was evident in the loop length of cancer nuclei denoting an increase in size

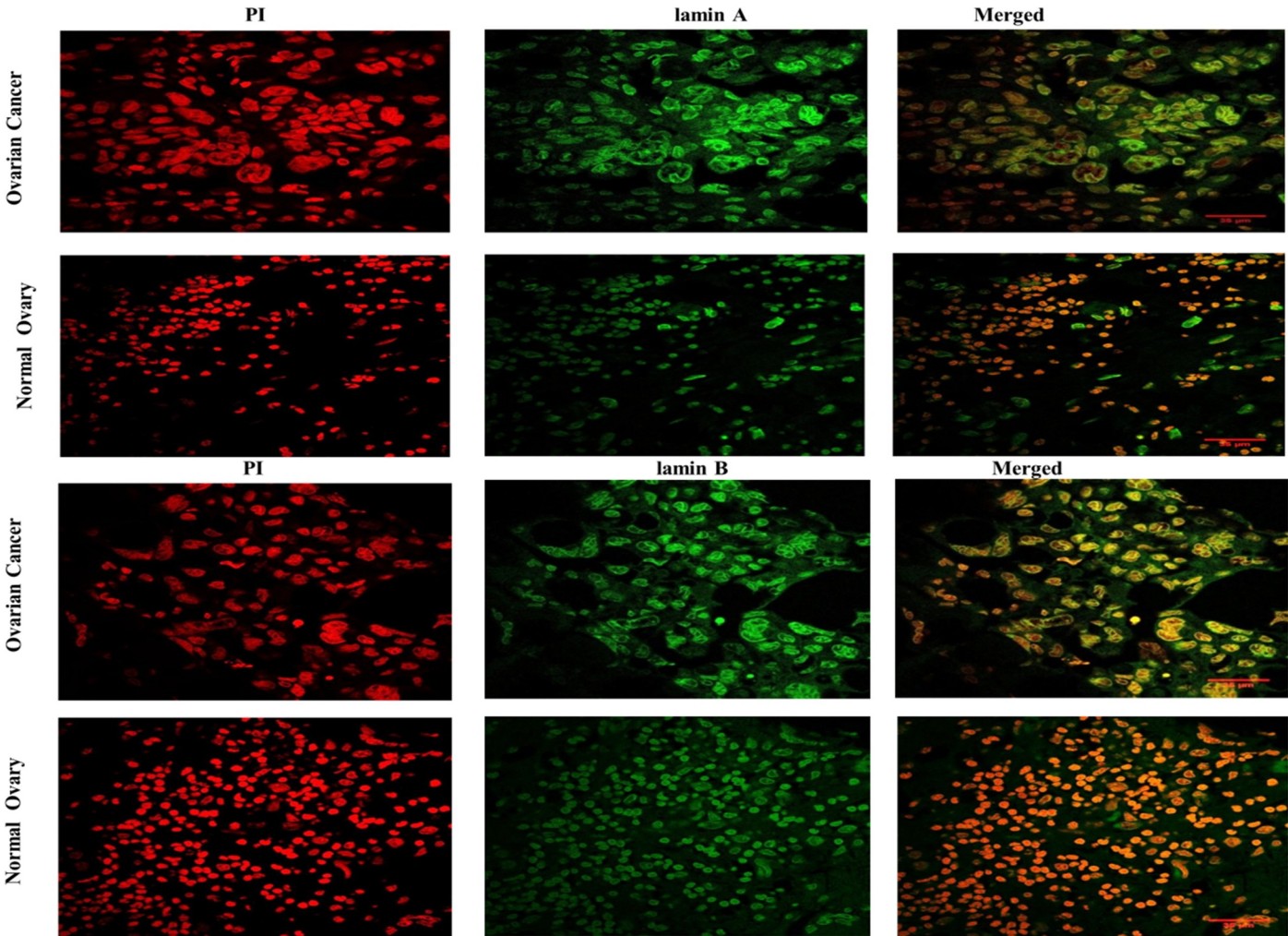

**Fig 2. Representative images of confocal micrographs showing the distribution of lamin A and lamin B in tissues from Ovarian cancer and Normal Ovary.**
Images of Ovarian Cancer and Normal ovarian tissue nuclei have been marked in their respective columns. Propidium Iodide staining of the nuclei is shown in the first panel containing red channel images. Lamin A and lamin B distributions respectively have been shown in the second panel of green channel images. Merged images of both channels have been shown in the third panel. Magnification: 63X. Scale Bar: 35 μm.

once again (**S2D1 and S2D2 Fig**). Next, to study the change in the surface architecture, maximum curvature and normalized curvature were measured; but no significant shift was observed to deduce a conclusion (**S2E1, S2E2, S2F1, S2F2 Fig**). As we all know, that tumor microenvironment harbors a heterogeneous cell population including cells at different stages of malignancy and some normal cells too, so the analysis spanned a large range of parametric measures to account for all the nuclei in the population. Overall, these measurements confirmed prominent alteration in morphology in the cancer nuclei or the nuclei approaching malignancy with respect to the normal nuclei and gave a gross idea regarding the direction of change. This experiment concluded that morphometric alteration in form of altered distribution of lamins in nuclei could potentially be used as signatures to classify cancer and normal nuclei or to study the progress towards malignancy.

Each parameter from the morphometric dataset of cancer and normal lamin A and lamin B-stained nuclei have been concatenated and plotted as histograms to evaluate the distribution pattern. A normal distribution is obtained upon Standard Scaling. Each of the morphometric

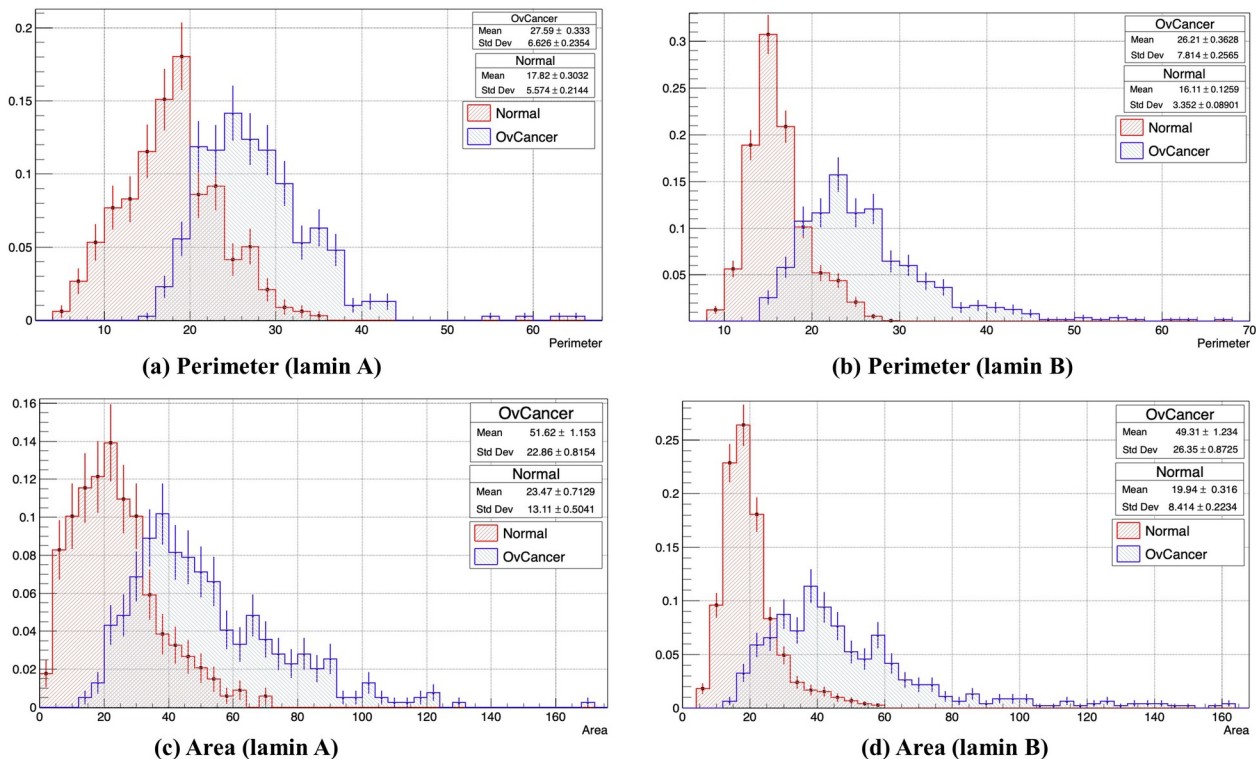

**(a) Perimeter (lamin A)**

**(b) Perimeter (lamin B)**

**(c) Area (lamin A)**

**(d) Area (lamin B)**

**Fig 3. Histograms showing distributions of the normal and ovarian cancer nuclei based on different morphometric parameters obtained from lamin A and B stained tissue sample images.** The X-axis denotes the normalized number of nuclei with respect to the total number of nuclei calculated. Y-axis denotes the measure of the parameter. (a) Comparative distribution of the number of normal (Mean±Std error of mean:17.82± 0.3032) (Std Dev:5.574±0.2144) and ovarian cancer (Mean±Std error of mean:27.59± 0.333) (Std Dev:6.626±0.2354) nuclei based on Perimeter values acquired from lamin A-stained tissues. (b) Comparative distribution of the number of normal (Mean±Std error of mean:16.11± 0.1259) (Std Dev:3.352±0.08) and ovarian cancer (Mean±Std error of mean:26.21± 0.3628) (Std Dev:7.814±0.2565) nuclei based on Perimeter values acquired from lamin B-stained tissues. (c) Comparative distribution of the number of normal (Mean±Std error of mean:23.47± 0.7129) (Std Dev:13.11 ±0.0541) and ovarian cancer (Mean±Std error of mean:51.62±1.153) (Std Dev:22.86±0.8153) nuclei based on Area values acquired from lamin A-stained tissues. (d) Comparative distribution of the number of normal (Mean±Std error of mean:19.94± 0.316) (Std Dev:8.414±0.2234) and diseased (Mean±Std error of mean:49.31± 1.234) (Std Dev:26.35±0.8725) nuclei based on Area values acquired from lamin B-stained tissues.

parameters used in the study is following approximate normal distribution (**S3 Fig**). As the area was the most important feature among the eight morphometric parameters as determined by the gini index, it was chosen to be the output variable. A Correlation Matrix was generated to analyze how the output variable is correlated with the other parameters. In order to better represent the data on the cells with multidimensional attributes, we used PCA (Principal component analysis) [53] such that the correlated dimensions are automatically removed (**S4 Fig**). Three classifiers (Adaptive Boosting Classifier, Random Forest Classifier, and Decision tree Classifier) with 5-fold cross-validation were used to generate probability scores. Accuracy from AdaBoost, Decision Tree, and Random Forest classifiers were 90%, 80%, and close to 90% respectively. Probability scores from the Adaptive Boosting Classifier were used as final outputs from Pipeline 1.

## Data augmentation and pre-processing

We started our experiment with around 50,000 ovarian cancer and normal nuclei. The pre-processing algorithm consisted of two parts–applying a segmentation mask based on the key visual properties like area, perimeter, circularity, eccentricity, foci distance, loop length,

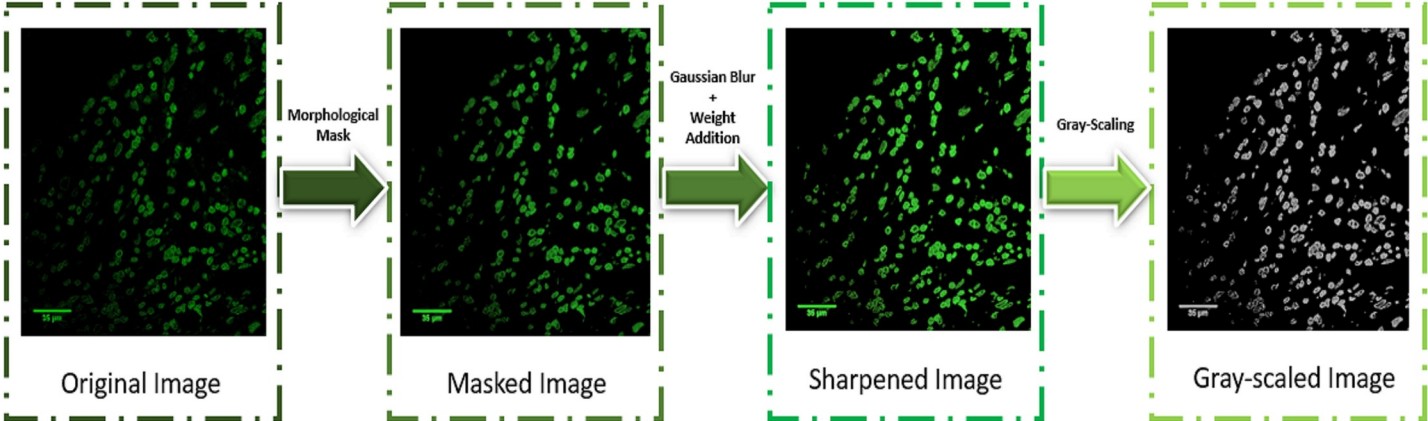

**Fig 4. Sequence of operations followed for pre-processing of the dataset.** In the first part, a segmentation mask was applied based on the morphometric properties followed by image sharpening to have the key features more visually prominent. In the second part, the sharpened image, which was obtained using Gaussian Blurring and weight addition, was transformed to its greyscale version to improve the computational time of the model.

maximum curvature, and normalized curvature of the nuclei followed by image sharpening techniques. In the first part, based on the Image Hue Saturation Value (HSV) and using a sensitivity factor, the segmentation mask was created, which was subsequently made prominent by the application of morphological closing operation with an elliptical kernel. The elliptical kernel was used to adapt to the shape of the nuclei and capture the maximum possible relevant information. The rectangular kernel was previously tested and it got no more than 93% accuracy. In the second phase, using Gaussian Blur and adding weights to the blurred image, we ensured uniform sharpening of the pre-processed images with the segmentation masks and converted the pre-processed images into a grey-scaled form so that the key visual features were rendered more prominent and easier for the Deep Learning algorithm to unravel features. Information from the background was completely removed to emphasize the morphological properties of the nuclei. (**Fig 4**) The significance of pre-processing is highlighted by the fact that the raw images used as training dataset resulted in only 71% accuracy which significantly improved upon pre-processing.

## Training a deep hybrid learner and validation

After the pre-processed images were acquired, we had to split the data into a training set and validation set with a split ratio of 75:25. The training set was used to train the supervised binary classification model and the validation set (Using 5-folds cross-validation techniques) was used for hyper-parameter tuning to make sure that the model was not over-fitting on the training set and remained generalized. For training a Deep Hybrid Learner, we first trained the 21 Layered CNN which was used to extract features. We trained it for 250 epochs with a learning rate of 0.00025 and a batch size of 32. We have used Adam [54] as the optimizer and cross-entropy loss as the loss function. We have elucidated the Model accuracy score, Model PR score, Model AUC-ROC score, and Precision score recall score as the matrices for determining the fitness of the algorithm (**Fig 5A–5F**). We observed from the learning curves that the training and validation AUC Scores gradually increased with training iterations or epochs and the maximum training score after 250 epochs were obtained as 0.998 and the validation score was obtained as 0.997. The model loss curve for the training and validation set gradually decreased with an increase in epoch (**Fig 5F**), which was an indication that the model was learning gradually with more training iterations. The absence of any statistically significant variance between training

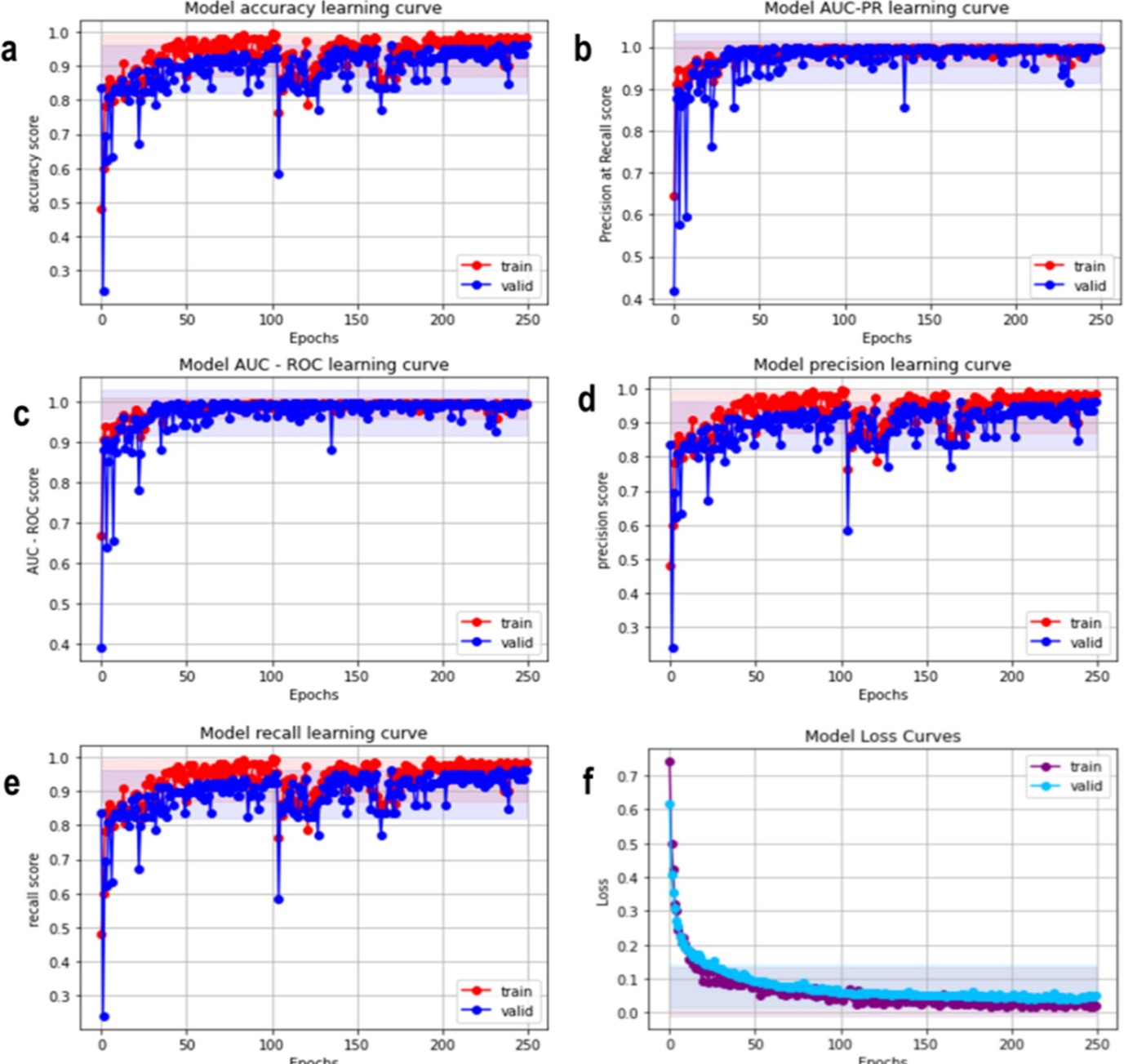

**Fig 5.** Model evaluation matrices: a. Model accuracy learning curve. b. Model AUC-PR learning curve c. Model AUC-ROC learning curve. d. Model precision learning curve. e. Model recall learning curve f. Model Loss curve.

and validation loss indicated the absence of any over-fitting issues with very minimal false positives and false negatives shown in AUC-ROC and AUC–PR plots (**Fig 6A–6D**)

## Model evaluation on test data

We compared performances of deep hybrid learning models with other models as mentioned previously in the material and method section. In the transfer learning models, we used pre-

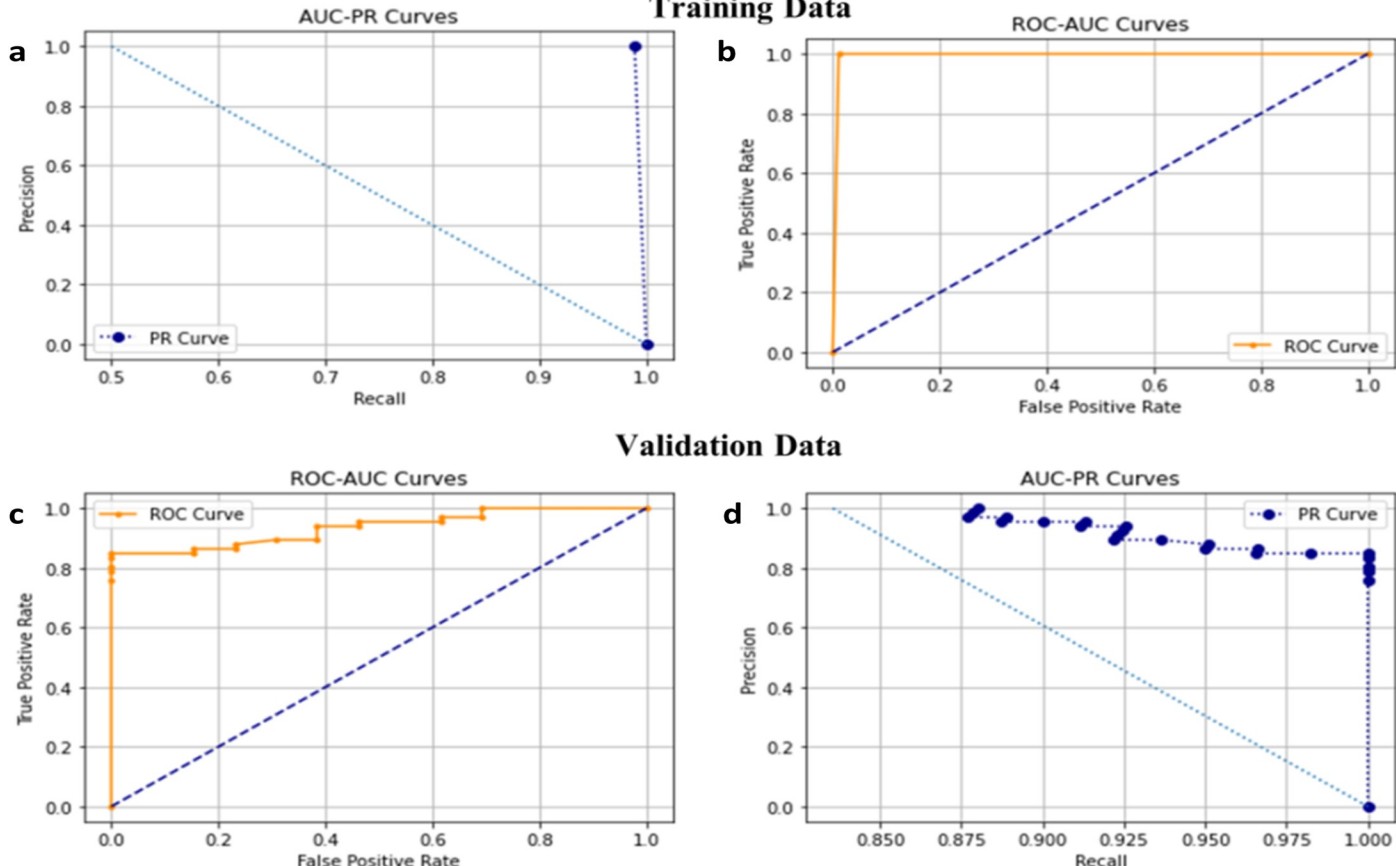

**Fig 6.** a. ROC-PR curve on Training set b. AUC-ROC curve on the Training set. c. AUC-ROC curve on the validation set. d. AUC-PR curve on the validation set.

trained weights from ImageNet. The training, validation, and test dataset were the same for all the approaches and epoch numbers, batch size was also consistent for all the approaches. Test images were unknown to the model and the clinical details were not revealed to the person performing the tests to ensure an unbiased validation and impartial selection of the accurate model based on performance. Deep hybrid learners were found to be the best working models for this specific problem (**Figs 7 and 8**). For this research work, the choice of the ideal model architecture depended on two main criteria: Generalization and Efficiency. From the above results we could see that the Deep Hybrid Learners (both Random Forest and XGBoost variant) showed almost consistent results for training, validation and testing phases. Also, we found that the model was extremely efficient with low variance, as we observed that the AUC scores on training validation and test dataset were 0.99, 0.99, and 1.0 respectively (**Fig 8**). The conventional deep learning model trained from scratch without transfer learning seemed to have high training scores, but it showed high variance on validation and test datasets as the scores were much lower on validation and test set. Therefore, it indicated that the model was overfitting on the training data, and it was not generalized, hence performing poorly on the testing and validation dataset. This behavior of the model could be explained by our previous hypothesis that the dataset used for this research work was not favorable for a conventional deep learning approach, as it would require more training samples for the conventional model to learn and improve generalization. Hence, more sophisticated and novel approaches like Deep Hybrid Learning which uses CNN for feature extraction and classical machine learning

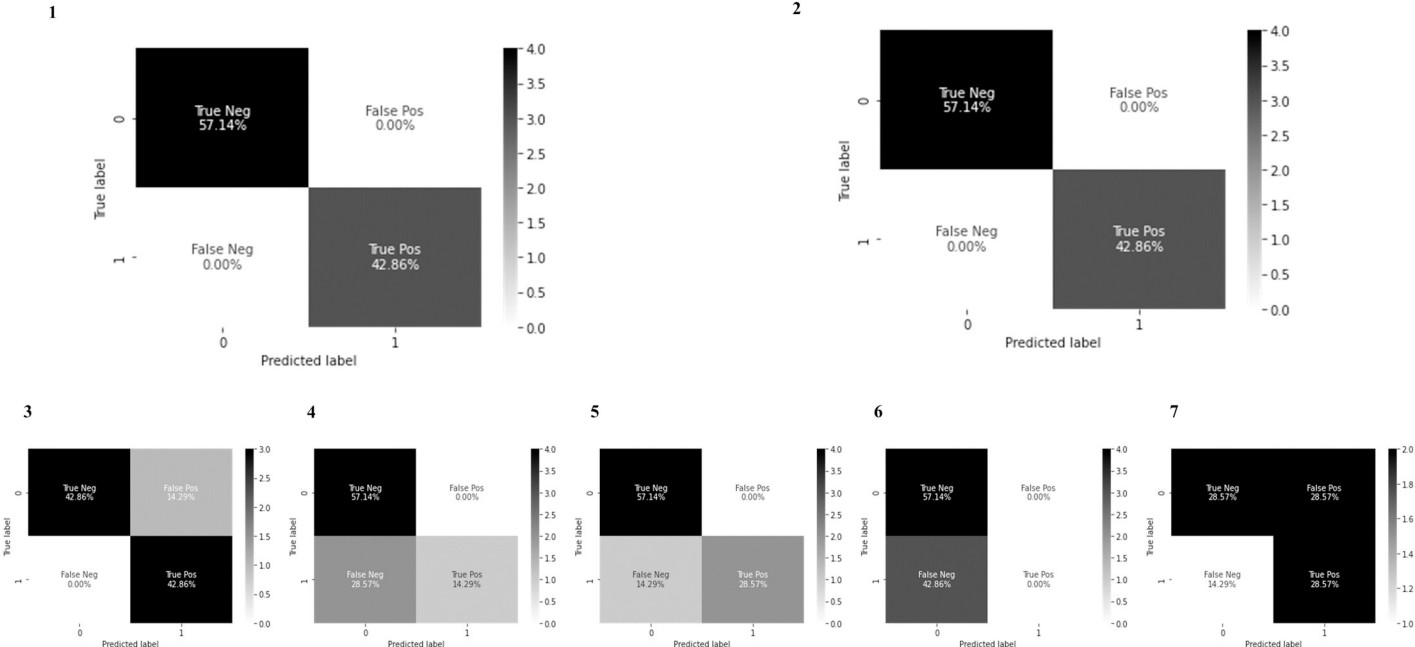

**Fig 7. Confusion matrices.** In this research work we have compared the Deep Hybrid Learning with both XGBoost and Random Forest variant, with a conventional Deep Neural Network (without transfer learning and having the same 21 layered CNN as DHL), DenseNet201 with transfer learning, InceptionNet v3 with transfer learning, ResNet50 with transfer learning and VGG16 with transfer learning. 'Normal' and 'Cancer' has been denoted as 0 and 1 respectively in the confusion matrices. **1.** Deep Hybrid Learning with Random Forest **2.** Deep Hybrid Learning with XGBoost **3.** Conventional Deep Neural Network Model **4.** DenseNet201 with transfer learning **5.** ResNet50 with transfer learning **6.** InceptionNetv3 with transfer learning **7.** VGG16 with transfer learning.

| Model Architecture | Training AUC Score | Validation AUC Score | Testing AUC Score |
|---|---|---|---|
| Deep Hybrid Learner with Random Forest | 0.99 | 0.99 | 1.0 |
| Deep Hybrid Learner with XGBoost | 0.99 | 0.99 | 1.0 |
| Conventional Deep Neural Network (without Transfer Learning) | 0.99 | 0.88 | 0.88 |
| DenseNet201 with Transfer Learner | 0.91 | 0.70 | 0.67 |
| ResNet50 with Transfer Learning | 0.98 | 0.52 | 0.83 |
| InceptionNet v3 with Transfer Learning | 0.50 | 0.50 | 0.50 |
| VGG16 with Transfer Learning | 0.96 | 0.81 | 0.58 |

**Fig 8. Comparison of model evaluation matrices.**

algorithms for the final classification, were more efficient and robust for this type of micro-scopic image dataset. We have even applied Transfer Learning [49] with more sophisticated Deep Learning architectures like DenseNet201 [48], ResNet50, InceptionNetv3, VGG16 [51], but the results obtained showed the presence of over-fitting, lack of generalization, and much lower model efficiency than the Deep Hybrid Learners. One plausible reason could be that these transfer learning models were trained using pre-trained weights from ImageNet images, which were significantly different and might have a significant statistical difference from microscopic images, making the transfer learning approach ineffective in this case. Thus we can conclude that our Deep Hybrid learning approach was successful and much better per-forming than other deep learning algorithms with these types of microscopic image datasets for automated detection of ovarian cancer.

## Deep hybrid learning

The trained and hyper-parameter tuned model performance on the test set proved how well the model was generalized and did not have any unwanted bias. Now, as an imbalanced dataset is not suitable for classical deep learning models for building supervised classifiers with high accuracy and generalization, we came up with the Deep Hybrid Learning (DHL) algorithm, which utilized Deep Convolutional Neural Network to extract features from the pre-processed samples and uses the extracted feature vector with classical Machine Learning algorithms like Random Forest [55] and XGBoost [33] to build the final classifier. Of late Ensemble learning techniques like Boosting algorithms are known to work well with high dimensional data, as boosting techniques are known to combine weak learners to identify the "hard" data points and combine the weak learners to form a very strong and efficient classifier [56]. Similarly, Ensemble methods like Random Forests work very well on smaller but high dimensional data-sets for solving binary classification problems and have been known to produce generalized results [55]. The results obtained using the Deep Hybrid Learning approach turned out to be extremely promising (**Fig 8**) and so far, have performed much better than any other conven-tional approaches and are comparable to or even better than human-level performance for this classification problem.

## Summary and conclusion

We have demonstrated nuclear A and B-type lamins as diagnostic markers for ovarian cancer thereby modulating nuclear morphology. In other words, their altered expression or distribu-tion acts as a function of nuclear shape and size and has been used as a tool to detect and diag-nose malignancy in the context of ovarian cancer. Confocal images of tissue samples elucidated a significant increase in area and perimeter in the ovarian cancer nuclei which is in good agreement with the fact that the cell nuclei in the ovarian tumor tissues are mostly associ-ated with an enlargement in size compared to the cell nuclei of normal tissues. Progressively, we increased the sample size and attempted to evaluate the possibility of quantitative feature extraction of nuclei and characterization by projecting nuclear morphology as a potential tool to distinguish normal and ovarian cancer tissues by introducing a novel deep hybrid learning network. We first focussed on extracting the pattern of the images and then moving a step fur-ther, noise reduction was performed so that it becomes convenient for the model to distinguish between cancer and normal tissue images. Pre-processing the images using various advanced mathematical and morphological techniques, like sharpening, masking, smoothening, etc. enhanced the differentiating pattern of the images so that the neural model could easily iden-tify them with utmost precision. This was followed by advanced techniques of data augmenta-tion to create all sorts of simulated practical tests by training the deep neural network model

over the microscopic image dataset, using the Deep Hybrid learning approach, resulting in a much more reliable system than standard CNN.

## Findings and significance

But, it's a challenging problem to custom tailor the process of feature extraction to classify cancer from microscopic image datasets with fastness and precision. On the other hand, modern techniques of deep learning have already proven their superiority to perform image classification with minimal pre-processing and in many cases, it has outperformed domain experts in classifying images. Thus, we decided to use Deep Learning approaches based on CNN (Convolutional neural network) for our study. For this specific problem, we propose that the distribution of lamins and size of the nucleus are important factors to classify images with cancer and normal nuclei. Hence, we trained a deep learning network specifically crafted *denovo* for the particular challenge of classifying ovarian cancer. Different types of augmentation bring almost all possible kinds of representation of the images resulting in a much higher possibility that a real-time test image would be very similar if not the same and would be having a greater probability to be classified with perfection in real-time. Also, the dropout used here was 0.20 which is a regularization technique used for preventing the overfitting of models. We have introduced a combination of classical machine learning algorithms (XGBoost, SVM, Random Forest) with standard CNN and designed a state-of-the-art deep hybrid network. We created a completely automated pipeline architecture consisting of pipelines 1 and 2 where processing images to predicting cancer or non-cancer would be performed within seconds. The strength of our method lies in the fact that the model showed 99.8% training accuracy and 99.7% validation accuracy in distinguishing normal and ovarian cancer cell nuclei and with our feedback mechanism the network could be retrained with wrong predictions made to further improve the accuracy. The novelty of this method is the stringency of prediction, which is magnified due to the incorporation of morphometric parameters of the nuclei along with random pre-processed images. It is emphasized that our model outperformed other transfer learning models like ResNet50, AlexNet, VGG-16, DenseNet. Furthermore, the metric scores obtained were much better than other state-of-the-art Deep Learning architectures like DenseNet201 and even when the approach of transfer learning failed. One of the main reasons for Deep Hybrid Learning to be successful is because we had replaced the final fully-connected layers with a machine learning (typically ensemble learning) algorithm, which made the overall model efficient and generalized.

## Limitations

Nonetheless, we were restricted to limited sample size as a part of the pilot project. Furthermore, the samples were collected from one institution catering to the specific demography of patients. Although it sufficed our purpose, it would be desirable to increase the sample size as well as to adopt a multi-institutional approach for diversity in the future. Also, for the initial feature extraction, we had used a custom CNN. But based on the results obtained from the conducted experiments, we can conclude that our framework is robust and efficient and worked perfectly well within the tenets of the pilot project. So, it is certain that with sufficient images in the future this would outperform other models and will be able to classify not only between normal and cancer nuclei but will also be able to predict the degree of risks of benign nuclei to become cancerous.

## Current status

The strength and future research direction for using AI for cancer prediction and early diagnosis should be concentrated on cancers that currently do not have a clear natural history

identified and therefore a screening strategy. We plan to extend our work in larger datasets and especially in diverse chemotherapy response categories; especially supplementing chemotherapy response score (CRS) after neoadjuvant chemotherapy and interval debulking surgery, where the CRS2 score is the grey zone and requires better biomarkers for prognostic stratification. We predict a large scope for our approach of interpreting alterations in cellular architecture in the early detection and screening strategies in ovarian cancer. The majority of HGS (High grade serous) ovarian cancers arise from fallopian tubes; many women undergo fimbriectomy for sterilization purposes as well as a prophylactic measure for prevention of ovarian cancer in high-risk individuals i.e., BRCA mutations. It would be interesting to study whether alteration in the nuclear architecture in the fimbriae/ovary could be one of the early predictors for developing cancer. More importantly, this may be used as a prognostic marker in addition to the standard immunohistochemistry and histology in ovarian cancer if a clinical correlation can be shown in future studies and that remains one of our target research strategies in the future including application in other women cancers. A lot of studies are trying to detect precursor lesion signatures like STIC (serous tubal intraepithelial cancer) and p53 signature in ovarian/tubal cancer.

## Future scope

As a future scope, we would try to test the framework on larger sample size and in a multi-institutional setting. The ultimate goal would be to replace the custom CNN with custom transformer architecture to obtain more contextual information from the data and evaluate if the custom transformer architecture version is giving better results.

## Supporting information

**S1 Fig. Network diagram.** For the Incept Layer, it utilized the number of filters for the Conv2D sub-layer and another hyper-parameter for the number of filters for the Left and Right Conv2D sub-layer as input. In both the Conv2D sub-layers, after tuning, a learning rate of 0.1 and an activation function of Leaky ReLu were used to learn the non-linear relationships in the underlying high dimensional data. For the initial Conv2D layer, we have used a filter dimension of 5x5 and strided convolution with stride as 2. For Left Conv2D a filter size of 3x3 was used and for the Right Conv2D, a filter size of 5x5 was used. Finally, both the left and right conv2d sub-layers were concatenated and passed to the next layer. The Squeeze layer followed the same structure as Incept Layer. The learning rates and the activation function used in the sub-layers were the same, the only difference being with the filter dimensions. For the initial Conv2D sub-layer, the dimension was (1x1) with stride 1 while for Left Conv2D & Right Conv2D the filter dimensions were (1x1) and (3x3) respectively. Like the Incept layer, the Left and the Right Sub-layers were concatenated and passed to the next layers. After a series of Incept and Squeeze layers we used another Conv2D layer with 64 filters and each filter was of dimension 3x3, with an activation function of Leaky ReLu with a learning rate of 0.1. A combination of dropout and L2 regularization was used to reduce the chances of overfitting. Finally, after all the convolution layers which were used to extract the features, we flattened the output and passed the flattened output to classical Machine Learning algorithms like XGBoost and Random Forest for the final classification part.
(DOCX)

**S2 Fig. Histograms showing distributions of the normal and ovarian cancer nuclei based on different morphometric parameters obtained from lamin A and B stained tissue sample images before and after pre-processing.** The X-axis denotes the normalized number of nuclei

with respect to the total number of nuclei calculated. Y-axis denotes the measure of the parameter. A. 1. Comparative distribution of the number of normal (Mean±Std error of mean:0.8796 ± 0.005994) (Std Dev:0.1285±0.00495) and ovarian cancer (Mean±Std error of mean:0.8452± 0.006767) (Std Dev:0.1347±0.004785) nuclei based on Circularity values acquired from lamin A stained tissues. A. 2. Comparative distribution of the number of normal (Mean±Std error of mean:0.9309± 0.003013) (Std Dev:0.8024±0.002131) and ovarian cancer (Mean±Std error of mean:0.8974± 0.005269) (Std Dev:0.1135±0.003726) nuclei based on Circularity values acquired from lamin B stained tissues. B. 1. Comparative distribution of the number of normal (Mean±Std error of mean:0.7067± 0.01169) (Std Dev:0.2144±0.008269) and ovarian cancer (Mean±Std error of mean:0.7885± 0.008241) (StdDev:0.164±0.005827) nuclei based on Eccentricity values acquired from lamin A stained tissues. B. 2. Comparative distribution of the number of normal (Mean±Std error of mean:0.6348± 0.008105) (Std Dev:0.2147±0.005731) and diseased (Mean±Std error of mean:0.7076± 0.008841) (Std Dev:0.19±0.006252)nuclei based on Eccentricity values acquired from lamin B stained tissues. C. 1. Comparative distribution of the number of normal (Mean±Std error of mean:5.204± 0.1648) (Std Dev:3.03±0.1165) and ovarian cancer (Mean±Std error of mean:8.942± 0.1729) (Std Dev:3.441±0.1223) nuclei based on Foci Distance values acquired from lamin A stained tissues. C. 2. Comparative distribution of the number of normal (Mean±Std error of mean:3.934± 0.07391) (Std Dev:1.968 ±0.05226) and diseased (Mean±Std error of mean:7.414± 0.1738) (Std Dev:3.744±0.1229) nuclei based on Foci Distance values acquired from lamin B stained tissues. D. 1. Comparative distribution of the number of normal (Mean±Std error of mean:12.17± 0.3004) (Std Dev:5.523 ±0.2124) and ovarian cancer (Mean±Std error of mean:20.01± 0.3152) (Std Dev:6.282±0.2232) nuclei based on Loop Length values acquired from lamin A stained tissues. D. 2. Comparative distribution of the number of normal (Mean±Std error of mean:9.869± 0.1275) (Std Dev:3.396 ±0.09018)and diseased (Mean±Std error of mean:17.5± 0.3255) (Std Dev:7.011±0.2302)nuclei based on Loop Length values acquired from lamin B stained tissues. E. 1. Comparative distribution of the number of normal (Mean±Std error of mean:1.1015± 0.03876) (Std Dev:0.7105 ±0.02741) and ovarian cancer (Mean±Std error of mean:0.7776± 0.02854) (Std Dev:0.5651 ±0.02018) nuclei based on Maximum Curvature values acquired from lamin A stained tissues. E. 2. Comparative distribution of the number of normal (Mean±Std error of mean:0.797± 0.01755) (Std Dev:0.4672±0.01241)and diseased (Mean±Std error of mean:0.6361± 0.0222) (Std Dev:0.4782±0.01571)nuclei based on Maximum Curvature values acquired from lamin B stained tissues. F. 1. Comparative distribution of the number of normal (Mean±Std error of mean:0.5354± 0.01359) (Std Dev:0.2496±0.009609) and ovarian cancer (Mean±Std error of mean:0.4402± 0.01048) (Std Dev:0.2086±0.007414nuclei based on Normalized Curvature values acquired from lamin A stained tissues. F. 2. Comparative distribution of the number of normal (Mean±Std error of mean:0.6323± 0.008435) (Std Dev:0.2246±0.005964)and diseased (Mean±Std error of mean:0.5489± 0.01401) (Std Dev:0.2243±0.007362) nuclei based on Normalized Curvature values acquired from lamin B stained tissues.
(DOCX)

**S3 Fig. All parameters are showing Gaussian distribution: Eight parameters from the morphometry dataset of the lamin A and B stained ovarian cancer and normal nuclei have been concatenated and plotted as histograms and all of them are showing approximate Gaussian distribution.**
(DOCX)

**S4 Fig. Correlation matrix: Correlation matrix to analyze how the output variable is correlated with the other parameters.** PCA was used further to minimize correlation.
(DOCX)

**S1 Table. Fundamental image quality analysis.**
(DOCX)

**S2 Table. Model layers and dimensions.**
(DOCX)

**S1 Dataset.**
(TXT)

**S2 Dataset.**
(TXT)

**S3 Dataset.**
(TXT)

**S4 Dataset.**
(TXT)

## Acknowledgments

All tissues and TMA samples were obtained from Tata Medical Center, provided by Asima Mukhopadhyay, a clinician-scientist (Gynaecological Oncologist), based at Tata Medical Center and currently at Chittaranjan National Cancer Institute Kolkata and her research group. Biobanking for ovarian cancer tissues and TMA creation was performed through the DST-UKIERI grant held by Dr. Asima Mukhopadhyay. The histograms were generated using the ROOT data analysis framework with the help of Mr.Gourab Saha, SRF, High Energy Nuclear, and Particle Physics Division, SINP who helped in writing the programs in C++. Duhita Sengupta thanks DAE for the fellowship. Kaushik Sengupta thanks SERB, DST& BARD project of DAE, Govt. of India.

## Author Contributions

**Conceptualization:** Kaushik Sengupta.

**Data curation:** Kaushik Sengupta.

**Formal analysis:** Duhita Sengupta, Joy Mustafi.

**Investigation:** Duhita Sengupta, Sk Nishan Ali, Aditya Bhattacharya.

**Methodology:** Duhita Sengupta, Sk Nishan Ali, Aditya Bhattacharya, Asima Mukhopadhyay, Kaushik Sengupta.

**Project administration:** Duhita Sengupta, Sk Nishan Ali, Aditya Bhattacharya, Kaushik Sengupta.

**Resources:** Kaushik Sengupta.

**Software:** Sk Nishan Ali, Aditya Bhattacharya, Joy Mustafi.

**Supervision:** Kaushik Sengupta.

**Validation:** Duhita Sengupta, Sk Nishan Ali, Aditya Bhattacharya, Kaushik Sengupta.

**Visualization:** Duhita Sengupta, Sk Nishan Ali, Aditya Bhattacharya, Kaushik Sengupta.

**Writing – original draft:** Duhita Sengupta, Sk Nishan Ali, Kaushik Sengupta.

**Writing – review & editing:** Duhita Sengupta, Sk Nishan Ali, Aditya Bhattacharya, Asima Mukhopadhyay, Kaushik Sengupta.

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
