## [Decision Letter · Decision Letter 0]

9 Sep 2021

PONE-D-21-25346

A Deep Hybrid Learning pipeline for accurate diagnosis of Ovarian Cancer based on Nuclear Morphology

PLOS ONE

Dear Dr. SENGUPTA,

Thank you for submitting your manuscript to PLOS ONE. After careful consideration, we feel that it has merit but does not fully meet PLOS ONE’s publication criteria as it currently stands. Therefore, we invite you to submit a revised version of the manuscript that addresses the points raised during the review process.

We look forward to receiving your revised manuscript.

Kind regards,

Balachandran Krishnan, Ph.D

Academic Editor

PLOS ONE

Journal Requirements:

4.Please review your reference list to ensure that it is complete and correct. If you have cited papers that have been retracted, please include the rationale for doing so in the manuscript text, or remove these references and replace them with relevant current references. Any changes to the reference list should be mentioned in the rebuttal letter that accompanies your revised manuscript. If you need to cite a retracted article, indicate the article’s retracted status in the References list and also include a citation and full reference for the retraction notice.

Reviewers' comments:

Reviewer's Responses to Questions

**Comments to the Author**

1. Is the manuscript technically sound, and do the data support the conclusions?

Reviewer #1: Yes

2. Has the statistical analysis been performed appropriately and rigorously? 

Reviewer #1: Yes

3. Have the authors made all data underlying the findings in their manuscript fully available?

Reviewer #1: Yes

4. Is the manuscript presented in an intelligible fashion and written in standard English?

Reviewer #1: Yes

5. Review Comments to the Author

Reviewer #1: 1. Rephrase the sentence” But these verifications based on manual observations are never trustworthy”- in introduction. Human expert interventions are still trustworthy in many cases.

2. There is a repletion of the sentence “The minor and major axes of each nucleus were measured manually using ImageJ (ImageJ bundled with 64-bit Java 1.8.0_112)[39].” You can mention this only In the preprocessing session.

3. In page 6: “Here, we used PCA (Principal component analysis)[53] to minimize correlation”. The use of PCA is not to minimize correlation among attributes - It is to to describe correlations and reducing the dimensionality. Check this sentence or explain minimizing correlations in your context.

4. Section 3.5 has repeated statements that are already given under section 2.8.

5. Is the ratio 75:25 for training and testing or training and validation? There are contradicting statements in the article.

6. In fig 1 - The morphometric parameters are analyzed using ML model, Images are taken up by CNN what is the significance of an ML classifier again in pipeline 2?

7. Have you considered any other parameters like False positive rates which can be significant in case of these classifications. Since the dataset is not balanced, accuracy alone may not be a good evaluation.

8. It would be good if you could represent your model in the table with: Layer and dimension for a better understanding on the model.

9. Overall : The sentences in section 1 and 2 are very long and sometimes difficult if too many information are given in a single sentence. Try writing in short sentence.

10. Check for any repetitive sentences or concepts in the manuscript.

11. The figures given are not clear and the interpretations from figures are very difficult.

6. PLOS authors have the option to publish the peer review history of their article (what does this mean?). If published, this will include your full peer review and any attached files.

Reviewer #1: No

---

## [Author Response · Author response to Decision Letter 0]

14 Sep 2021

Response to Reviewers:

We thank the reviewerfor the constructive comments. We have addressed all the comments to the best of our knowledge 

Response to Reviewer #1:

Reviewer #1: 

1. Rephrase the sentence” But these verifications based on manual observations are never trustworthy”- in introduction. Human expert interventions are still trustworthy in many cases.

We thank the reviewer for this constructive comment. The sentence has been rephrased in line number 94-95 as follows,

“But these verifications based on manual observations are often cumbersome and sometimes prone to error.”

2. There is a repletion of the sentence “The minor and major axes of each nucleus were measured manually using ImageJ (ImageJ bundled with 64-bit Java 1.8.0_112)[39].” You can mention this only In the preprocessing session.

We thank the reviewer for this comment. The above mentionedsentence is there in line number 159 in the Material and Method section (Sub-section: Image analysis and data presentation) and the repeated line has been removed from line number 250 in the Result section (Sub-section: Morphometric classification, Standard Scaling, and Adaptive Boosting).

3. In page 6: “Here, we used PCA (Principal component analysis)[53] to minimize correlation”. The use of PCA is not to minimize correlation among attributes - It is to to describe correlations and reducing the dimensionality. Check this sentence or explain minimizing correlations in your context.

We thank the reviewer for this comment. We agree with the reviewer and we sincerely apologize for the inadvertent mistake in phrasing the sentences. We have rephrased the PCA part in the sentences between the lines 306-308.

“In order to better represent the data on the cells with multidimensional attributes, we used PCA (Principal component analysis) [53] such that the correlated dimensions are automatically removed”

We have observed correlated attributes [S4 Fig, S5 Fig] while training classical ML pipeline, thus, PCA had been used to not only reduce the dimensions of the data set but also to reduce the adverse effect of correlated attributes to improve robustness of the model. Because, having multicollinearitywithin the variables highly affects the variance associated with any problem, also can affect the interpretation of any model, as it undermines the statistical significance of independent variables.

4. Section 3.5 has repeated statements that are already given under section 2.8.

We thank the reviewer for this comment. We apologize for the mistake. As per the requirement of the format of PLOS ONE, the labels for section and Subsections have been removed. The statements which were mentioned in line number 223-227 in Material and Method section (Subsection: Comparison of DHL with other Deep learning approaches) have been removed from line number 385-390 in the Result section (Sub section: Model Evaluation on Test Data).

5. Is the ratio 75:25 for training and testing or training and validation? There are contradicting statements in the article.

We thank the reviewer for this constructive comment. In the text, the words “test” and “validation” were used in close context with respect to our model and could be read interchangeably.

6. In fig 1 - The morphometric parameters are analyzed using ML model, Images are taken up by CNN what is the significance of an ML classifier again in pipeline 2?

We thank the reviewer for this comment. Morphometric parameters (in tabular format) were used as inputs in pipeline 1and the images (after pre-processing) were used as inputs in pipeline 2. Since, the inputs were different in the two pipelines, ML classifier were used in both the pipelines separately to make the model more generalised and robust.

7. Have you considered any other parameters like False positive rates which can be significant in case of these classifications. Since the dataset is not balanced, accuracy alone may not be a good evaluation.

We thank the reviewer for this comment. We would like to point out that we have rigorously studied the true positive and false positives which are validated by AUC-ROC curve and AUC-PR curve for training data and the same for validation data which is very robust. Please refer to Fig 6 following an explanation already included between lines 367-376.

8. It would be good if you could represent your model in the table with: Layer and dimension for a better understanding on the model.

We thank the reviewer for this comment. We would like to draw the attention of the reviewer to S2 Fig in the supplementary where we had already provided a Network Diagram with the layers. However, with due respect to the reviewer and his/her comments, we have also provided a table this time with Layer and dimension information of the model as asked in Supplementary information section (S6 Table).

9. Overall: The sentences in section 1 and 2 are very long and sometimes difficult if too many information are given in a single sentence. Try writing in short sentence.

We thank the reviewer for this comment.We apologize for the same and we have rephrased long sentences into short and simplified ones in line numbers 72-75 in section 1 and line numbers 166-167 and 194-196 in section 2.

10. Check for any repetitive sentences or concepts in the manuscript.

We thank the reviewer for this comment. We sincerely apologize for the mistakes. We have checked meticulously and carefully eliminated all repeating sentences from the text from line numbers 258-159,385-390,261-264

11. The figures given are not clear and the interpretations from figures are very difficult.

We thank the reviewer for this comment. We have tried to convey the results obtained from our experiments in the figures with maximum information and utmost clarity followed by clear interpretations to the best of our knowledge. However, we apologize if the figures appeared complicated to the reviewer. We beg to approach the reviewer to consider our humble submission.

---

## [Decision Letter · Decision Letter 1]

25 Nov 2021

A Deep Hybrid Learning pipeline for accurate diagnosis of Ovarian Cancer based on Nuclear Morphology

PONE-D-21-25346R1

Dear Dr. SENGUPTA,

We’re pleased to inform you that your manuscript has been judged scientifically suitable for publication and will be formally accepted for publication once it meets all outstanding technical requirements.

Kind regards,

Balachandran Krishnan, Ph.D

Academic Editor

PLOS ONE

Additional Editor Comments (optional):

Reviewers' comments:

Reviewer's Responses to Questions

**Comments to the Author**

1. If the authors have adequately addressed your comments raised in a previous round of review and you feel that this manuscript is now acceptable for publication, you may indicate that here to bypass the “Comments to the Author” section, enter your conflict of interest statement in the “Confidential to Editor” section, and submit your "Accept" recommendation.

Reviewer #1: All comments have been addressed

2. Is the manuscript technically sound, and do the data support the conclusions?

Reviewer #1: Yes

3. Has the statistical analysis been performed appropriately and rigorously? 

Reviewer #1: N/A

4. Have the authors made all data underlying the findings in their manuscript fully available?

Reviewer #1: Yes

5. Is the manuscript presented in an intelligible fashion and written in standard English?

Reviewer #1: Yes

6. Review Comments to the Author

Reviewer #1: This work proposes a Deep Hybrid 42 Learning model, though derived from classical machine learning algorithms and standard 43 CNN, showed a training and validation AUC score of 0.99The review comments were addressed properly. The paper on modification looks good.

7. PLOS authors have the option to publish the peer review history of their article (what does this mean?). If published, this will include your full peer review and any attached files.

Reviewer #1: No

---

## [Editor Report · Acceptance letter]

30 Dec 2021

PONE-D-21-25346R1 

A Deep Hybrid Learning pipeline for accurate diagnosis of Ovarian Cancer based on Nuclear Morphology 

Dear Dr. Sengupta:

I'm pleased to inform you that your manuscript has been deemed suitable for publication in PLOS ONE. Congratulations! Your manuscript is now with our production department. 

Kind regards, 

on behalf of

Dr. Balachandran Krishnan 

Academic Editor

PLOS ONE